# Use of Biological Drugs for Psoriasis: A Drug-Utilization Study Using Tuscan Administrative Databanks

**DOI:** 10.3390/ijerph19116799

**Published:** 2022-06-02

**Authors:** Sabrina Giometto, Silvia Tillati, Laura Baglietto, Nicola De Bortoli, Marta Mosca, Marco Conte, Marco Tuccori, Rosa Gini, Ersilia Lucenteforte

**Affiliations:** 1Unit of Medical Statistics, Department of Clinical and Experimental Medicine, University of Pisa, 56126 Pisa, Italy; sabrina.giometto@med.unipi.it (S.G.); silvia.tillati@med.unipi.it (S.T.); laura.baglietto@unipi.it (L.B.); 2Unit of Gastroenterology, Department of Clinical and Experimental Medicine, University of Pisa, 56126 Pisa, Italy; nicola.debortoli@unipi.it; 3Unit of Rheumatology, Department of Clinical and Experimental Medicine, University of Pisa, 56126 Pisa, Italy; marta.mosca@unipi.it; 4CESP, U1018 INSERM, Université Paris-Saclay, UVSQ, Hôpital Paul Brousse Bâtiment 15/16, CEDEX, 94807 Villejuif, France; marconte89@gmail.com; 5Unit of Pharmacology and Pharmacovigilance, Department of Clinical and Experimental Medicine, University of Pisa, 56126 Pisa, Italy; marco.tuccori@gmail.com; 6Unit of Adverse Drug Reactions Monitoring, University Hospital of Pisa, 56126 Pisa, Italy; 7Tuscan Regional Healthcare Agency, 50141 Florence, Italy; rosa.gini@ars.toscana.it

**Keywords:** psoriasis, switching, biologics, drug utilization

## Abstract

Our study aims at providing evidence on patterns of use of biologic drugs for psoriasis in Tuscany, Italy. We conducted a drug-utilization study based on administrative databanks of Tuscany (EUPAS45365) from 2011 to 2019. We selected new users of etanercept, infliximab, adalimumab, ustekinumab, or secukinumab between 1 January 2011 and 31 December 2016. We considered subjects with psoriasis and followed subjects until the end of the study period (three years after the first dispensation of biologic drug for psoriasis) or the patient’s death, whichever came first. We censored subjects for pregnancy or neoplasia. For each subject, we defined the state as the weekly coverage of one of the biologic drugs of interest. We then defined the switch as the change from a state to another one. A total of 7062 subjects with a first dispensation of a PSObio drug in the inclusion period was identified, and 1839 (52.9% female, 51.6 mean age) patients were included in the analysis. Among new users of adalimumab (N = 770, 41.9%), one third showed a continuous behaviour whereas the others moved to etanercept and ustekinumab. New users of etanercept (N = 758, 41.2%), had the highest proportion of switchers, with adalimumab most often being the second choice. New users of infliximab (N = 159, 8.6%) experienced the highest proportion of treatment discontinuation. The present study suggests that the majority of patients treated with PSObio drugs do not switch from one active ingredient to another. However, patients who started biological therapy with etanercept had the highest frequency of switching to other PSObio drugs, whereas those who started with secukinumab or ustekinumab had the lowest.

## 1. Introduction

Psoriasis (PSO) is an inflammatory skin disease that encompasses different clinical phenotypes. In the last 30 years, there has been much progress in the comprehension of its pathogenesis, but there is currently no curative treatment for it [1].

The prevalence of PSO ranges from 0.1% in east Asia to 1.5% in western Europe, being more common among Caucasians and in high-income countries. It increases linearly with age, and it arises equally in males and females, with an earlier onset in females, at 16–22 years and 55–60 years. About 70–80% of patients present a mild condition that can be managed with topical treatment alone [1,2].

The severity of PSO is generally measured through clinical scales such as the Psoriasis Area and Severity Index (PASI), the body surface area (BSA), and the Dermatology Life Quality Index (DLQI), a questionnaire that assesses the impact of the disease on the quality of life [3]. However, there are no generally recognized categories of severity. According to European Guidelines, each national society should define its own method of PSO severity grading based on extension, location of lesions, degree of inflammation, responsiveness to treatment, and effect on quality of life [3]. A moderate-to-severe condition is defined for PASI > 10 or BSA > 10, and DLQI > 10, and a mild condition for PASI ≤ 10 and BSA ≤ 10 and DLQI ≤ 10. Treatment goals have been well defined to help physicians in their clinical practice, and they are based on both measures of severity of skin lesions (PASI, BSA) and impact on the quality of life (DLQI). Disease improvement is indicated as percentage of PASI change from baseline, with a PASI75 parameter that indicates the percentage of patients who had at least a 75% reduction of the baseline PASI score during pharmacological treatment. Although a PASI75 response is generally considered a clinically relevant improvement, the treatment aims at an almost complete remission of the disease, with a reduction of the baseline PASI of at least 90%. A treatment failure is defined as PASI50 not being achieved. The treatment goal should be assessed at the end of the induction period at 12 or 16 weeks, depending on the active principles, and every 8–12 weeks during the maintenance period.

According to the recent guidelines [3], treatment of PSO could be divided into different pathways depending on disease severity: mild conditions should be managed using conventional treatments such as topicals alone; in moderate-severe cases a systemic treatment is recommended, with psoralens and ultraviolet A (P-UVA) combinations [4,5,6,7], conventional drugs (methotrexate, cyclosporine and tacrolimus), or a first-line label biologic if a sufficient treatment success cannot be expected with conventional treatment. Systemic treatment with biologic drugs has become the approved standard in moderate-severe PSO, with or without PsA, to reduce not only lesion extension but also to curb the inflammation, blocking the inflammatory pathways with a more tailored therapy (compared to conventional drugs) and reducing the number of doses needed to obtain clinical improvements [8,9]. There are many families of systemic biologic drugs, and their classification is based on their molecular targets: anti-tumor necrosis factor (TNF) alpha (Adalimumab, Infliximab, Etanercept), anti-IL-12/23 (Ustekinumab), anti-IL-17 (Brodalumab, Ixekizumab, Secukinumab), and anti-IL-23 (Guselkumab, Risankizumab, Tildrakizumab) [10]. Etanercept, infliximab, and ustekinumab are indicated as second-line treatments, whereas the others are first-line treatments among biologic drugs [3]. Most of these drugs are also available in the management of other inflammatory cutaneous conditions, such as hidradenitis suppurativa [11]. The reasons for discontinuing the first biological or switching to another one might be primary or secondary failure of treatment, intolerance, or occurrence of adverse events. Despite the frequency of switches in clinical practice, it is unclear which drugs are most frequently switched to, and no clinical recommendations are available.

Therefore, our study aims at providing evidence on patterns of use of biologic drugs for psoriasis in Tuscany, Italy. To the best of our knowledge, it is the first study that aims at investigating all switches in clinical practice, from any PSObio drug to any PSObio drug.

## 2. Materials and Methods

We conducted a drug-utilization study based on administrative databanks of Tuscany, a region in central Italy.

The study protocol was published in the European Network of Centres for Pharmacoepidemiology and Pharmacovigilance (ENCePP^®^) registry (EUPAS45365).

Administrative databanks contain longitudinal pseudonymized subject-level information on the utilization of healthcare services reimbursed by the National Healthcare Service and dispensed to all subjects who are residents and registered with a general practitioner in the region. Each databank is linked to others at the patient level. 

We used the following databanks: the inhabitant registry, including information on gender, date of birth, and subject’s date of entry and of exit;the drug dispensing registry, including information on date, type, dose and number of packages of dispensed drugs by community or hospital pharmacies to subjects; drug is coded according to the Anatomical Therapeutic Chemical classification system (ATC);the exemption from co-payment registry, containing the release date and disease of subjects; exemption is coded according to the Italian exemption code;the hospital discharge registry, including information on date of hospital admission, date of hospital discharge, primary and secondary diagnoses and procedures of subjects admitted to the hospital; diagnosis is coded according to the ICD-9-CM;the emergency department (ED) registry, containing information on date of ED admission, date of ED discharge, primary and secondary diagnoses of subjects admitted to ED; diagnosis is coded according to the ICD-9-CM;the outpatient services registry, collecting date of visit/test, record of specialist encounters, without diagnosis code, and diagnostic tests without results of subjects attending visits;the certificates of childbirth assistance, terminations, and miscarriages records, including information on duration of gestation, date of delivery, duration of amenorrhea, date of termination of pregnancy.

### 2.1. Study Population and Cohort Definition

We selected subjects with a first dispensation of etanercept, infliximab, adalimumab, ustekinumab, or secukinumab (related ATC codes: L04AB01, L04AB02, L04AB04, L04AC05, L04AC10) between 1 January 2011 and 31 December 2016 (inclusion period). The date of the first dispensation was defined as index date (ID), and the drug of the first dispensation was defined as index drug. We adopted a new-user approach by selecting only subjects without dispensations of biological drugs for psoriasis (PSObio drug) in the year preceding the ID (look-back period). From the cohort of new users of PSObio drugs, we excluded subjects not resident in Tuscany and those with a look-back period shorter than 365 days. We considered subjects with a diagnosis of psoriasis (i.e., ICD-9-CM code 696.1 in the hospital discharge or the ED registry) or an exemption for psoriasis (Italian exemption code 045 in the exemption from co-payment registry) five years before or one year after the ID or a dermatological visit (codes 89.7, 89.01 and delivery specialty code 052) one year before or one year after the ID. We included only subjects with three years of follow-up after the ID.

### 2.2. Follow-Up

All subjects accumulated person time from the ID until the end of study period (three years after ID), or patient’s death, whichever came first.

We censored subjects for pregnancy and hospitalizations or accesses to emergency department (ICD-9 codes: 140*–239*) or exemptions (Italian exemptions code: 048*) related to neoplasia.

### 2.3. Study Period

The study was conducted from 2011 to 2019, with six years of inclusion period (2011–2016) and three years of follow-up.

### 2.4. Variable of Interest: Switch 

For each subject, we defined the state as the weekly coverage of one of the following drugs: etanercept, adalimumab, infliximab, ustekinumab and secukinumab. The absence of coverage of one of the drugs listed above was also considered as a state. We followed patients for three years; therefore, 156 states were calculated for each subject. We defined the switch as the change from a state to another one.

### 2.5. Covariates

We considered the following characteristics: demographic variables (age and gender), comorbidities (lung disease, myocardial infarction, stroke, hypertension, other cardiovascular (CV) disease, diabetes, hip/spine/leg fracture, depression, gastrointestinal ulcer, other gastrointestinal disorders, Sjögren’s syndrome, rheumatoid nodules, myopathies, polyneuropathy, cancer, and concomitant therapies (glucocorticoid for systemic use, non-steroidal anti-inflammatory drugs, NSAIDs, opioid and non-opioid analgesics, and small-molecule drugs for psoriasis, see Appendix A for codes).

### 2.6. Data Analysis

Alluvial plots were used to illustrate the flows of switch, from the index date (time 1), every twelve weeks until the end of the observation period (time 14). Each time was given the prevalent state over the 12-week period.

We then performed a state sequence analysis, i.e., a cluster analysis based on the hierarchical agglomerative method within each index drug group, in order to better visualize the different pattern of switch. We grouped similar longitudinal patterns of switch within each group who started the biological treatment with the same index drug. Since available statistical procedures to determine the optimal number of clusters is partially dependent on the order of data, a plausibility criterion was used. We reported graphically all results per single index drug. The first graph of each figure describes the state of each subject in the three years of observation: the length of the segments represents the time spent in that state, considering the week as the unit of time; the colour, as per the legend, indicates the type of PSObio drug, whereas grey indicates no PSObio drug. The subsequent graphs in each figure show, instead, the weekly cross-sectional state at each time unit over the three-year observation period; the ordinate represents the frequency of the different states per time unit. We then calculated the number of patients with no, one, or two or more switches for each index drug, to better explore the switching behaviour.

Data analysis was performed using R, version 4.0.5, and the TraMineR package [12] was used to perform the state sequence analysis.

## 3. Results

A total of 7062 subjects with a first dispensation of PSObio drug in the inclusion period was identified (Figure 1). 

Among these subjects, 1450 were not included in the population registry of Tuscany, and 159 had a look-back period shorter than 365 days. A cohort of 5453 subjects was identified. One thousand eight hundred and eighty subjects were included because they had a diagnosis of PSO five years before or one year after ID (262 from hospital discharge, 12 from ED registry, 946 from exemption) or a dermatologic visit within one year before or after ID (1165). We excluded 41 subjects because they had less than three years of follow-up; thus, 1839 subjects were included in the study.

Table 1 summarizes the main characteristics of the cohort.

At ID, 973 subjects (52.9%) were female, and mean age was 51.6 (SD 15.2). The index PSObio drug most frequently reported was adalimumab (770, 41.9%), followed by etanercept (758 patients, 41.2%), infliximab (159, 8.6%), ustekinumab (115, 6.3%), and secukinumab (37, 2.0%).

In the year before ID, the mean number of dermatological visits was 3.7 (SD 3.7). Subjects had mainly a history of hypertension (59, 3.2%), cancer (39, 2.1%), diabetes (36, 2.0%), and other cardiovascular diseases (37, 2.0%), but most of patients did not have any of the comorbidities considered (1652, 89.8%). Most patients used at least one non-biologic drug for psoriasis treatment (1148, 62.5%) or systemic drug (981, 53.3%). Concomitant therapies most frequently used were: nonsteroidal anti-inflammatory drugs (NSAIDs) (968, 52.7%) and glucocorticoids (963, 52.4%). 

The majority of patients reached the end of the study (1766, 96.0%,); the remaining were censored for cancer (33, 1.8%), death 20 (1.1%), or pregnancy (19, 1.0%).

Figure 2 shows the flows of switches, from the index date (time 1), every twelve weeks until the end of the observation period (time 14).

The total number of patients using etanercept, infliximab or adalimumab decreased over time; the number of patients using ustekinumab remained constant; secukinumab users and those not covered by any PSObio drug increased over time. The use of ixekizumab was minimal and only observed in the third year of follow-up of a few patients whose index year was 2015 or 2016.

Figure 3, Figure 4 and Figure 5 panels A show state sequences plots of the group of individuals starting biological therapy with adalimumab, etanercept and infliximab, respectively; panels B, C and D represent the trend over time of the three main clusters of switches identified.

The graphs following the first one in each figure should be read cross-sectionally: for example, in the first week of the third quarter of cluster 2 of Figure 4, it is shown that 62.5% of subjects had a dispensation of the index drug, 5.1% of ustekinumab, 3.1% of secukinumab, 0.9% of adalimumab, and finally, 28.4% were not covered by any drug of interest.

For new users of adalimumab (N = 770, 41.9%, Figure 3), three clusters of switches were identified: the first including individuals switching to a state of no coverage (cluster 1, Figure 3B); the second including those who changed medication over the three years of follow-up, mainly switching to etanercept or ustekinumab (cluster 2, Figure 3C); the last one including one third of the adalimumab new users showing a continuous behaviour (cluster 3, Figure 3D).

For the group of patients who started the biological therapy with etanercept (N = 758, 41.2%, Figure 4A), three clusters were identified: the first including patients who largely switched to adalimumab (cluster 1, Figure 4B); the second including patients with a continuous behaviour (cluster 2, Figure 4C); the last one including patients who switched to a state of no coverage (cluster 3, Figure 4D), which was most often the second choice (cluster 1, Figure 4B).

Patients who initiated the biological treatment with infliximab (N = 159, 8.6%, Figure 5A) experienced the highest proportion of treatment discontinuation (cluster 3, Figure 5D). Of those changing biological therapy, most switched to adalimumab or, to a lesser extent, to etanercept (cluster 2, Figure 5C).

Infliximab was the drug least frequently switched to, regardless of the index drug, whereas adalimumab was the drug most frequently switched to by new users of etanercept and infliximab.

There were few subjects who initiated biological treatment with secukinumab (N = 37, 2%, Appendix A), and among them one patient switched to etanercept, one to ustekinumab, and one first to adalimumab and then to ustekinumab. New users of ustekinumab (N = 115, 6.3%, Appendix A) had the highest percentage of patients continuing therapy with the index drug and tended not to change medication (cluster 3, Appendix A).

The distribution of the number of switches to another PSObio drug overall and among the five groups defined by the index drug is reported in Table 2.

The majority of the subjects did not have any switch to another PSObio drug (73.1%); among the five index-drug groups, the higher frequency of one switch was in the new users of infliximab (23.3%), and the higher frequency of two or more switches was in the new users of etanercept (8.0%) (Table 2). The mean time to first switch was 224 days (SD 219) for secukinumab, 391 days for etanercept (SD 305), 409 days for infliximab (SD 278), 454 days for adalimumab (SD 296), and 485 days for ustekinumab (SD 276).

## 4. Discussion

The present study suggests that the majority of patients treated with PSObio drugs do not switch from one active ingredient to another. The highest proportion of switches to another PSObio drug was observed in patients who started biological therapy with etanercept. Patients with secukinumab or ustekinumab as index drug had the lowest frequency of switch to other PSObio drugs; however, it should be noted that these groups are also the least numerous, as these drugs were approved for psoriasis later than anti-TNF drugs. In the state sequence analysis, we observed that infliximab and etanercept new-users groups had the largest cluster of discontinuers, i.e., the one with a large grey area, while the ustekinumab new users had the largest continuers cluster, i.e., the one with a large area of the colour of the index drug. 

Our results seem to be in line with what has already been observed in the literature. Several studies observed higher adherence in patients treated with ustekinumab and lower adherence in those treated with etanercept [13,14,15,16]. Moreover, it was observed that patients treated with etanercept had a higher percentage of switches to other biologics, whereas patients treated with ustekinumab had the lowest percentage of switches [13]. This may be due to the different frequency of administration in the maintenance phase: that of etanercept is weekly (even biweekly during the first three months of treatment), while that of ustekinumab is quarterly [17]. The largest cluster of discontinuers observed for patients with infliximab as index drug could be due to route of administration: it is likely that infliximab is administered in the acute phase and then it is possible to switch to another treatment with an easier route of administration. It is also possible that discontinuation of infliximab or switching from infliximab to another treatment is due to tolerability; however, our data does not permit testing any hypothesis on safety.

We found that the active principle most frequently switched to by new users of anti-TNF was adalimumab, whereas new users of ustekinumab switched more frequently to secukinumab and vice-versa. Switching from an anti-TNF to a second anti-TNF may be effective, in particular, in cases of secondary failure or intolerance, whereas the efficacy of the second anti-TNF seems to be lower in cases where the reason for the switch was a primary failure [18]. In general, a suboptimal response to one anti-TNF does not predict the response to any other anti-TNF [19]. A good response to adalimumab was observed in patients previously treated with other biological drugs for psoriasis, including other anti-TNF drugs [19,20,21,22,23]. Moreover, patients previously treated with etanercept were then successfully treated with infliximab [24], and patients who did not respond to secukinumab showed an improvement in clinical parameters with ustekinumab [25,26] and adalimumab [26].

Considering the overall percentage of patients who had at least one switch, we found higher values than reported by other studies. An observational study conducted on the Italian Psocare Registry [18] recorded 5% of patients who had at least one switch between 2005 and 2010, over a follow-up period of three years. Another observational study conducted in California between 2009 and 2012 observed an overall percentage of switchers of 8% over one year of observation period [13]. Over a period of observation of three years, we found an overall percentage of switchers of 27%, but compared to Doshi et al. [13], we considered a longer observation period and we included secukinumab in the analysis, as it was approved during the study period. Piaserico et al. [18] considered an observation period of three years, as in our study, but they included only anti-TNF in the analysis, because ustekinumab and secukinumab had not yet been approved in the inclusion period of that study. Moreover, Piaserico et al. may have underestimated the percentage of switchers due to the type of data used, which was collected ad hoc by questionnaire; our study, based on administrative data, detects any dispensation not privately dispensed to the patient.

Looking at the patterns of use represented in the alluvial plot, we found that the use of anti-TNFs decreases over time, while that of ustekinumab remains constant and that of secukinumab increases.

This can be explained by the fact that secukinumab was approved for psoriasis in 2016, i.e., at the end of the inclusion period considered in our study.

Finally, we observe that the proportion of patients not covered by any PSObio drug increases over time.

The guidelines recommend [3] the use of biologics as a second-line treatment, after conventional DMARDs. Nevertheless, adalimumab and secukinumab might be recommended as a first-line treatment if a sufficient response from conventional systemic drugs is not expected, whereas etanercept, infliximab, and ustekinumab are recommended as second-line treatments. Despite this, in our study we observed that only 53.4% of the patients included in the cohort had used any conventional systemic drug for psoriasis in the year prior to the first biologic. Restricting the observation to new users of etanercept, infliximab, and ustekinumab, the situation did not change, and it was observed that only 55.8% of patients had used a conventional systemic before their first biologic. Some possible interpretations are that some patients may have received previous dispensations in regions other than Tuscany, which we could not take into account in this study; furthermore, some dispensations may have been made privately, as the cost of some conventional systemic drugs such as acitretin, cyclosporine and methotrexate is not as high as that of biologicals.

The first point of strength of our study consists in having observed a large cohort of patients and having considered all the switches that occurred in the follow-up period. Indeed, many studies in the literature focus on a specific switch, from etanercept to infliximab [24,27,28], from etanercept to adalimumab [22,23,29,30], etc., and are based on data collected in a single hospital; larger studies that consider all switches, from any PSObio drug to any PSObio drug, usually only consider the first switch for each patient [31,32].

Our study, being population-based, provided information on an unselected cohort of patients. Finally, we included Tuscan databanks, and Tuscany can be considered representative of the Italian population.

Our study also has some limitations. First, the therapeutic coverage was calculated using the DDD drug specific, not capturing changes in dosage or frequency of administration due to medical advice. Second, indications for use of the PSObio drugs considered were not available in the administrative databanks, and three of the five PSObio drugs considered have indications other than psoriasis; we addressed this problem by choosing an inclusion criterion of diagnosis of PSO or record of exemption or record of dermatologic visit, although this does not fully protect against including new users of biologicals for rheumatological reasons who also had psoriasis, but perhaps in a mild form. However, the above algorithm has not been validated for psoriasis. Third, we do not have the reason for the switch, as the results of the specialist visits are not recorded in the administrative databanks. However, it has been observed that lack of effectiveness is the main cause for switching to another biological, whereas switching due to safety reasons is less frequent [32,33,34,35].

## 5. Conclusions

The present study suggests that the majority of patients treated with PSObio drugs do not switch from one active ingredient to another. However, patients who started biological therapy with etanercept had the highest frequency of switching to other PSObio drugs. In contrast, patients with secukinumab or ustekinumab as the index drug had the lowest frequency of switching to other PSObio drugs; however, it should be noted that these groups are also the least numerous, as these drugs were approved for psoriasis later than anti-TNF drugs. Important in this respect is the different frequency of administration in the maintenance phase: that of etanercept is weekly, or even biweekly during the first three months of treatment, whereas that of ustekinumab is quarterly. Furthermore, we observed that the active ingredient most frequently switched to by new-users of anti-TNF was adalimumab, whereas new-users of ustekinumab switched more frequently to secukinumab and vice versa. The use of anti-TNFs decreased over time, while that of ustekinumab remained constant and that of secukinumab increased. Moreover, we observed that the proportion of patients not covered by any PSObio drug increased over time.

## Figures and Tables

**Figure 1 ijerph-19-06799-f001:**
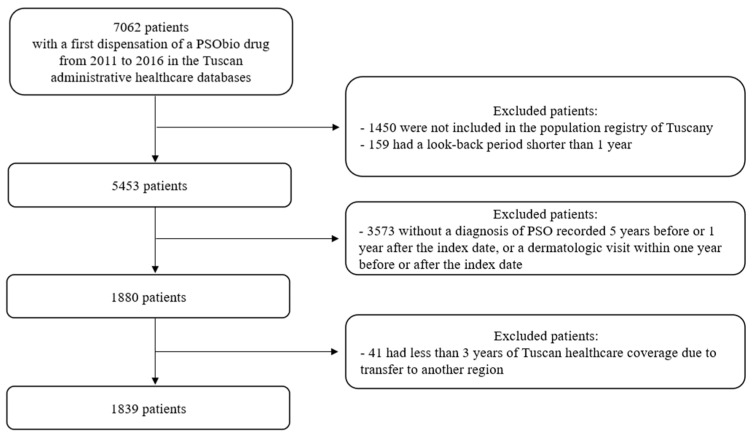
Study flow chart.

**Figure 2 ijerph-19-06799-f002:**
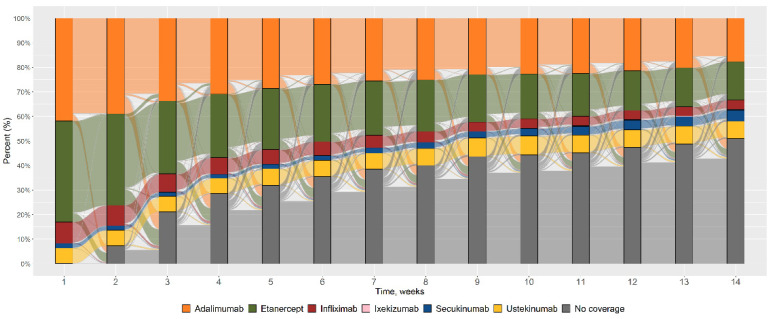
Pattern of utilisation as a percentage of the PSObio drugs considered, from the index date (t1) until the end of the three years of observation (t14).

**Figure 3 ijerph-19-06799-f003:**
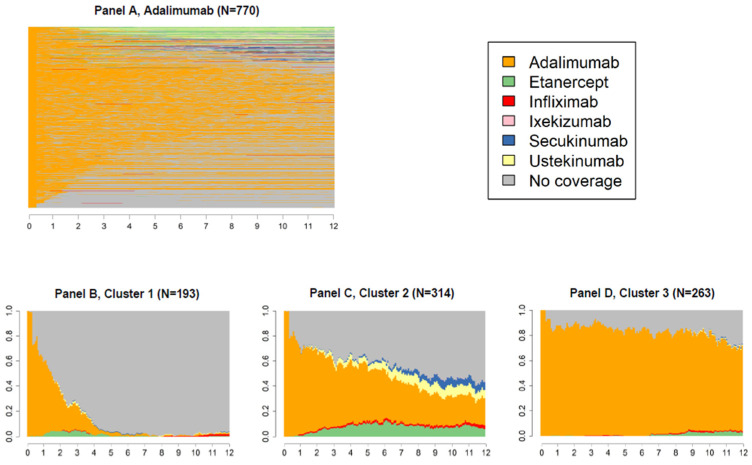
State sequences plots of the group starting biological therapy with adalimumab (N = 770, 41.9%), overall, and by cluster.

**Figure 4 ijerph-19-06799-f004:**
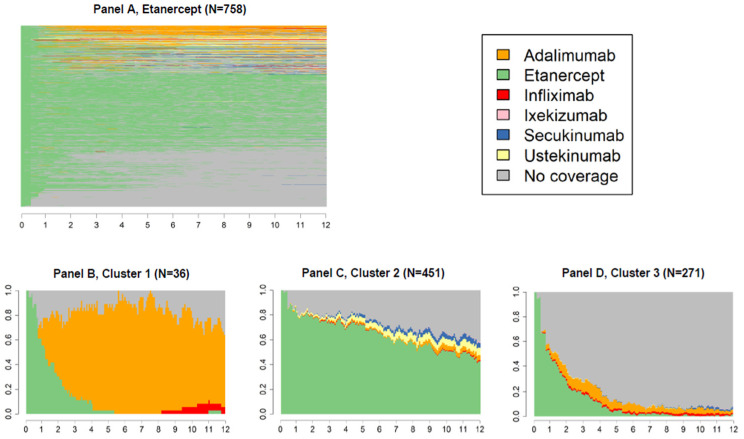
State sequences plots of the group starting biological therapy with etanercept (N = 758, 41.2%), overall, and by cluster.

**Figure 5 ijerph-19-06799-f005:**
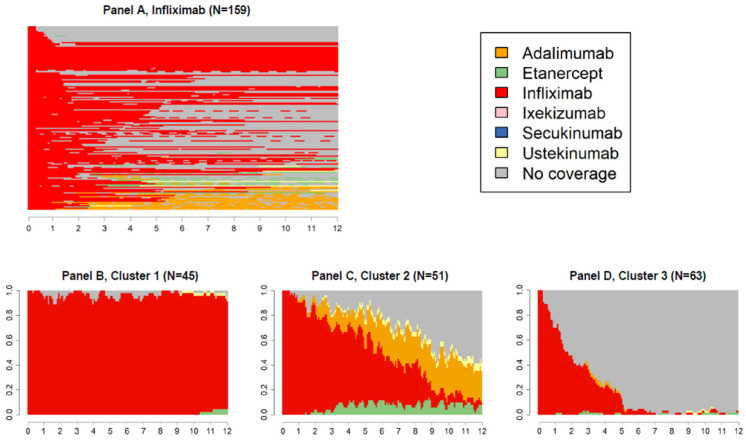
State sequences plots of the group starting biological therapy with infliximab (N = 159, 8.6%), overall, and by cluster.

**Table 1 ijerph-19-06799-t001:** Characteristics of 1839 subjects included in the cohort at index date (age, gender, and index drug) or one year before ID (visits, comorbidities, and concomitant therapies).

	N (%)
**At index date**	
*Gender*	
Female	973 (52.9)
Male	866 (47.1)
*Age*	
Mean (SD)	51.6 (15.2)
*Class of ages*	
0–20	61 (3.3)
21–40	352 (19.1)
41–50	421 (22.9)
51–60	443 (24.1)
61–70	375 (20.4)
71–80	167 (9.1)
81–100	20 (1.1)
*Index drug*	
Adalimumab	770 (41.9)
Etanercept	758 (41.2)
Infliximab	159 (8.6)
Secukinumab	37 (2.0)
Ustekinumab	115 (6.3)
**One year before ID (look-back period)**	
*Dermatologic visits*	
Mean (SD)	3.7 (3.7)
Median (1° quartile–3° quartile)	2 (1–5)
*Comorbidities*	
Lung disease	24 (1.3)
Myocardial infarction	4 (0.2)
Other CV disease	37 (2.0)
Stroke	11 (0.6)
Hypertension	59 (3.2)
Diabetes	36 (2.0)
Fracture (of hip/spine/leg)	16 (0.9)
Depression	8 (0.4)
Gastrointestinal ulcer	0 (0.0)
Other gastrointestinal disorders	8 (0.4)
Sjögren’s syndrome	1 (0.1)
Rheumatoid nodules	0 (0.0)
Rheumatoid lung disease	4 (0.2)
Myopathies	1 (0.1)
Polyneuropathy	0 (0.0)
Cancer	39 (2.1)
None ^1^	1652 (89.8)
*Concomitant therapies*	
Non-biological drugs for psoriasis	1148 (62.5)
Acitretin	113 (6.1)
Anti-psoriatic for topical use	480 (26.1)
Apremilast	0 (0.0)
Cyclosporin	276 (15.0)
Methotrexate	713 (38.8)
Psoralens for systemic use	0 (0.0)
Psoralens for topical use	0 (0.0)
Retinoids for treatment of psoriasis	113 (6.1)
At least one systemic treatment	981 (53.3)
None ^2^	690 (37.5)
Glucocorticoid for systemic use	963 (52.4)
Non-Steroidal Anti-Inflammatory Drugs (NSAIDs)	968 (52.7)
Opioid analgesics	391 (21.3)
None ^3^	173 (9.4)

^1^ none of the above comorbidities ^2^ none of the previous non-biological drugs for psoriasis ^3^ none of the above among: non-biologic psoriasis drugs, glucocorticoids, non-steroidal anti-inflammatory drugs, opioid analgesics, and non-opioid analgesics.

**Table 2 ijerph-19-06799-t002:** Distribution of number of switchers to another PSObio drug overall and among the five groups defined by the index drug.

	Overall (N = 1839)	Etanercept (N = 758)	Adalimumab (N = 770)	Infliximab (N = 159)	Ustekinumab (N = 115)	Secukinumab (N = 37)
No switch, n (%)	1345 (73.1)	527 (69.5)	575 (74.7)	115 (72.3)	94 (81.7)	34 (91.9)
One switch, n (%)	378 (20.6)	170 (22.4)	153 (19.9)	37 (23.3)	16 (13.9)	2 (5.4)
Two or more switches, n (%)	116 (6.3)	61 (8.0)	42 (5.5)	7 (4.4)	5 (4.3)	1 (2.7)

## Data Availability

The data presented in this study are available on request from the corresponding author.

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
