# Peer review of "Use of Biological Drugs for Psoriasis: A Drug-Utilization Study Using Tuscan Administrative Databanks"

_ijerph, 2022, doi:10.3390/ijerph19116799_

Round 1

Reviewer 1 Report

An interesting drug utilization study based on an Italian database, of over 7000 patients treated with biological drugs, with almost 2000 included in the analysis. The main problem of this study is the fact that the period analyzed ended in 2016, so the conclusions drawn by this study, where only a limited number of biologics were present, are not applicable nowadays, as a wider number of drugs are available, and patients treated with anti TNF alpha are a lower percentage; still, I found the article interesting, and eligible to publication after minor assessments:

lines 67-71 " Systemic treatment with biologic drugs has become the approved standard, in moderate-severe PSO, with or without PsA, to reduce not only lesion extension but also to curb the inflammation, blocking the inflammatory pathways with a more tailored therapy (compared to conventional drugs) and reducing the number of doses needed to obtain clinical improvements. " this paragraph needs some references, such as: doi: 10.3390/healthcare9050543. and doi: 10.3390/pharmaceutics14020294.

line 77 you should add: "Most of these drugs are also available in the management of other inflammatory cutaneous conditions, such as hidradenitis suppurativa" and cite: doi: 10.3390/ijms21228436

Author Response

An interesting drug utilization study based on an Italian database, of over 7000 patients treated with biological drugs, with almost 2000 included in the analysis. The main problem of this study is the fact that the period analyzed ended in 2016, so the conclusions drawn by this study, where only a limited number of biologics were present, are not applicable nowadays, as a wider number of drugs are available, and patients treated with anti TNF alpha are a lower percentage; still, I found the article interesting, and eligible to publication after minor assessments:

Reply: As also replied to Reviewer 2, the study period was from 2011 to 2019. Six years of inclusion (2011-2016) and three years of follow-up. The patient entered the last day of 2016 was followed until the end of 2019, the patients entered the first day of 2011 was followed until the first day of 2014. Length of observation was the same for all patients and this protects against time-related bias. We added a paragraph ‘Study period’ in the Methods section (lines 142-144) and we improved the abstract (lines: 19-20).

lines 67-71 " Systemic treatment with biologic drugs has become the approved standard, in moderate-severe PSO, with or without PsA, to reduce not only lesion extension but also to curb the inflammation, blocking the inflammatory pathways with a more tailored therapy (compared to conventional drugs) and reducing the number of doses needed to obtain clinical improvements. " this paragraph needs some references, such as: doi: 10.3390/healthcare9050543. and doi: 10.3390/pharmaceutics14020294.

Reply: We thank the reviewer for the suggestion and have added the indicated references.

line 77 you should add: "Most of these drugs are also available in the management of other inflammatory cutaneous conditions, such as hidradenitis suppurativa" and cite: doi: 10.3390/ijms21228436

Reply: We thank the reviewer and have added the indicated sentence (lines: 79-81) and reference.

Reviewer 2 Report

Dear Authors,

Thank you very much for the possibility to read such an interesting paper. To the best of my knowledge this is the first world data study on switches from one to other biologics. I have some concerns that need to be adressed.

  1. Please improve Figures. It is almost impossible to read the description of the axis as well as the legends. It is supposed to be much bigger (mostly for figure 2)
  2. I would like to know why you did not include anti IL 17 (guselkumab and ixekizumab were approved in 2016 if I am not wrong). Is it because you did not find any data or just not included the drugs in the analysis?
  3. I would like to also know if possible what was the mean time + SD of the drug administration before the switch.
  4. I find it quite weird for secukinumab. In our experience patients treated for more than a year with cosentyx often present secondary failure to respond to treatment and are often transfered for anti-23 or 17.
  5. I would like to see the similar explanation (as you did for the etanercept) for infliximab. Why patients do discontinue the treatment? I believe it is mostly due to the i.v. injection and the lack of possibility to reintroduce the treatment due to alergic reactions
  6. There is also another aspect - as you earlier mentioned, there was an introduction of new drugs during the period you have studied. Do you think that doctors would encourage patients to change the medication for a better one (as proven in clinical trials)?
  7. Why the study included only periods of 2011-2016? 5 Years have passed, more drugs were introduced and probably new challanges appeared that are not included into the manuscript. My question is why to publish 5 year old data?

Author Response

Dear Authors,

Thank you very much for the possibility to read such an interesting paper. To the best of my knowledge this is the first world data study on switches from one to other biologics. I have some concerns that need to be adressed.

Please improve Figures. It is almost impossible to read the description of the axis as well as the legends. It is supposed to be much bigger (mostly for figure 2)

Reply: We thank the reviewer for the comment. We improved Figure 2 increasing the text size in the legend and axes.

I would like to know why you did not include anti IL 17 (guselkumab and ixekizumab were approved in 2016 if I am not wrong). Is it because you did not find any data or just not included the drugs in the analysis?

Reply: We had included these drugs in the analysis, but had not found any of them as an index drug, between 2011 and 2016. However, thanks to your suggestion, we also included these drugs in the three years of follow-up after inclusion and found some dispensations of ixekizumab in 12 patients out of 1839 included in the cohort. No substantial change in the results was observed. However, we have updated the graphs (Figure 2, Figure 3, Figure 4, Figure 5, Figure A1, and Figure A2) and text (Table 2, lines: 219-220, 239-242) in accordance with this change.

I would like to also know if possible what was the mean time + SD of the drug administration before the switch.

Reply: Thank you for your suggestion. We added the following sentence in the Results section (lines: 272-274): “The mean time to first switch was 224 days (SD 219) for secukinumab, 391 days for etanercept (SD 305), 409 days for infliximab (SD 278), 454 days for adalimumab (SD 296), and 485 days for ustekinumab (SD 276)”.

I find it quite weird for secukinumab. In our experience patients treated for more than a year with cosentyx often present secondary failure to respond to treatment and are often transfered for anti-23 or 17.

Reply: As only 37 patients started biological therapy with secukinumab, it is difficult to draw any conclusion about this group. We have specified this limitation in the discussion (lines: 280-282) and in the conclusion (lines: 375-377).

I would like to see the similar explanation (as you did for the etanercept) for infliximab. Why patients do discontinue the treatment? I believe it is mostly due to the i.v. injection and the lack of possibility to reintroduce the treatment due to alergic reactions

Reply: We agree with the Reviewer, low adherence to infliximab could be due to route of administration and tolerability. However, our study was not designed to answer safety questions. We added the following  sentences in the Discussion (lines 293-299): “The largest cluster of discontinuers observed for patients with infliximab as index drug could be due to route of administration: it is likely that infliximab is administered in the acute phase and then it is possible a switching to another treatment with an easier route of administration. It is also possible that discontinuation of infliximab or switching from infliximab to another treatment is due to tolerability, however, our data does not allow to test any safety hypothesis.”

There is also another aspect - as you earlier mentioned, there was an introduction of new drugs during the period you have studied. Do you think that doctors would encourage patients to change the medication for a better one (as proven in clinical trials)?

Reply: In Italy the use of novel drug is closely monitored by the health authority to verify appropriateness, mainly for sustainability reasons. It is very unlikely that clinicians could have switched patients in absence of lack of effectiveness or safety issues.

Why the study included only periods of 2011-2016? 5 Years have passed, more drugs were introduced and probably new challanges appeared that are not included into the manuscript. My question is why to publish 5 year old data?

Reply: The study period was from 2011 to 2019. Six years of inclusion (2011-2016) and three years of follow-up. The patient entered the last day of 2016 was followed until the end of 2019, the patients entered the first day of 2011 was followed until the first day of 2014. Length of observation was the same for all patients and this protects against time-related bias. We added a paragraph ‘Study period’ in the Methods section (lines 142-144) and we improved the abstract (lines: 19-20). 

Round 2

Reviewer 2 Report

Thank you for introducing changes. I recommend accepti the manuscript in its current versions